# Tracing animal genomic evolution with the chromosomal-level assembly of the freshwater sponge *Ephydatia muelleri*

Nathan J. Kenny [1,13,15✉], Warren R. Francis [2,15], Ramón E. Rivera-Vicéns [3], Ksenia Juravel [3], Alex de Mendoza [4,5,14], Cristina Díez-Vives [1], Ryan Lister [4,5], Luis A. Bezares-Calderón [6], Lauren Grombacher [7], Maša Roller [8], Lael D. Barlow [7], Sara Camilli [9], Joseph F. Ryan [10], Gert Wörheide [3,11,12], April L. Hill [9], Ana Riesgo [1,15] & Sally P. Leys [7✉]

The genomes of non-bilaterian metazoans are key to understanding the molecular basis of early animal evolution. However, a full comprehension of how animal-specific traits, such as nervous systems, arose is hindered by the scarcity and fragmented nature of genomes from key taxa, such as Porifera. *Ephydatia muelleri* is a freshwater sponge found across the northern hemisphere. Here, we present its 326 Mb genome, assembled to high contiguity (N50: 9.88 Mb) with 23 chromosomes on 24 scaffolds. Our analyses reveal a metazoan-typical genome architecture, with highly shared synteny across Metazoa, and suggest that adaptation to the extreme temperatures and conditions found in freshwater often involves gene duplication. The pancontinental distribution and ready laboratory culture of *E. muelleri* make this a highly practical model system which, with RNAseq, DNA methylation and bacterial amplicon data spanning its development and range, allows exploration of genomic changes both within sponges and in early animal evolution.

[1] Department of Life Sciences, The Natural History Museum, Cromwell Rd, London SW7 5BD, UK. [2] Department of Biology, University of Southern Denmark, Odense, Denmark. [3] Department of Earth and Environmental Sciences, Paleontology & Geobiology, Ludwig-Maximilians-Universität München, Richard-Wagner-Str. 10, 80333 München, Germany. [4] ARC Centre of Excellence in Plant Energy Biology, School of Molecular Sciences, The University of Western Australia, Perth, WA 6009, Australia. [5] Harry Perkins Institute of Medical Research, Perth, WA 6009, Australia. [6] College of Life and Environmental Sciences, University of Exeter, Stocker Rd, Exeter EX4 4QD, UK. [7] Department of Biological Sciences, University of Alberta, Edmonton, AB T6G 2E9, Canada. [8] European Molecular Biology Laboratory, European Bioinformatics Institute, Wellcome Genome Campus, Cambridge CB10 1SD, UK. [9] Department of Biology, Bates College, Lewiston, ME 04240, USA. [10] Whitney Lab for Marine Bioscience and the Department of Biology, University of Florida, St. Augustine, FL 32080, USA. [11] SNSB-Bayerische Staatssammlung für Paläontologie und Geologie, Richard-Wagner-Str. 10, 80333 München, Germany. [12] GeoBio-Center, Ludwig-Maximilians-Universität München, Richard-Wagner-Str. 10, 80333 München, Germany. [13] Present address: Faculty of Health and Life Sciences, Oxford Brookes, Oxford OX3 0BP, UK. [14] Present address: School of Biological and Chemical Sciences, Queen Mary University of London, Mile End Road, London E1 4NS, UK. [15] These authors contributed equally: Nathan J. Kenny, Warren R. Francis, Ana Riesgo. ✉email: nathanjameskenny@gmail.com; sleys@ualberta.ca

One of the key events in the history of life was the evolutionary transition from unicellular organisms to multicellular individuals in which differentiated cell types work cooperatively[1]. In animals, the events that enabled this transformation can often be inferred by comparing the genomes of living representatives of non-bilaterian animals to those of bilaterians and their sister taxa, and determining shared characters and key differences between them[2]. However, the origins of several fundamental metazoan traits, such as tissues and nervous systems, are still unknown. Determining the origin of these characteristics requires more robust and contiguous genomic resources than are currently available for non-bilaterian animal taxa.

Porifera, commonly known as sponges, are one of the first lineages to have evolved during the rise of multicellular animals[3] and are an essential reference group for comparative studies. The benchmark genome for sponges, *Amphimedon queenslandica*[4], has provided a wealth of insight into the genomic biology of sponges[5,6], yet studies of other sponge species have suggested that traits in *A. queenslandica* may not be representative of the phylum as a whole[7,8]. For example, its genome is one of the smallest measured in sponges[6], it is highly methylated in comparison to other animals[9], may have undergone some gene loss even in well-conserved families[10] and has been described as possessing an 'intermediate' genomic state, between those of choanoflagellates and metazoans[5]. There are over 9200 species of sponge (http://www.marinespecies.org/porifera/,[11]), and understanding whether the unusual characteristics of *A. queenslandica* are typical of this large and diverse phylum can only be tested with additional, and more contiguous, genome assemblies.

Sponges diverged from the metazoan stem lineage in the Neoproterozoic[12] and therefore are central to understanding the processes and mechanisms involved in the initial metazoan radiation. Sponges possess the fundamental characteristics shared by all animals, including development through embryogenesis to form tissues and signalling to coordinate whole body behaviour[13,14]. Most also have a highly conserved body plan, consisting of canals and pumping cells that filter water effectively[15]. However, within the four classes of sponges (Hexactinellida, Demospongiae, Calcarea, and Homoscleromorpha), several groups differ from this Bauplan. Glass sponges (Hexactinellida) have syncytial tissues and are the only sponges shown to propagate electrical signals[16] while Cladorhizida (Demospongiae) are carnivorous and capture crustaceans with hook-like spicules[17]. Despite the diversity of sponges in the most species-rich class, the Demospongiae, only one group, the Spongillida, made the transition to freshwater some 250–300 million years ago (Mya) (Fig. 1a), later diversifying into the extant range of modern taxa worldwide around 15–30 Mya[18].

The transition to freshwater is one of the most remarkable evolutionary trajectories marine animals can undergo, as it requires a complete spectrum of physiological adaptations to novel habitats. Not only can freshwater sponges, which consist of a single layer of cells over a scant extracellular matrix, control their osmolarity in freshwater[19], they can withstand extremely cold temperatures and even freezing, as they inhabit some lakes that see temperatures below −40 °C[20], and can also tolerate extreme heat and desiccation in desert sand dunes and high up on tree trunks[21,22]. Freshwater sponges are both unfamiliar and yet so common worldwide that under the right conditions they can foul drinking water reservoirs, waste treatment plants, intake pipes, and cooling systems for power plants[23]. The main adaptation which permits colonisation of such extreme habitats is the production of sophisticated structures called gemmules, a distinctive stage in the life of these sponges[21]. The events that allowed colonisation of freshwater which are required for adaptation to extreme habitats by sponges are not yet fully understood[24]. Whether genomes of freshwater species are remarkably changed from those seen in marine sponges and other animals is also yet to be investigated.

The freshwater sponge *Ephydatia muelleri* (Lieberkühn, 1856) (Fig. 1) is found in rivers and lakes throughout the northern hemisphere (Supplementary Note 1, Supplementary Table 1). Because of its global distribution and century-long history of study both in situ and in the laboratory, *E. muelleri* is an outstanding model for asking questions about adaptation and the evolution of animal characters. It has separate sexes, allowing the study of inheritance[25], but more practically, gemmules are clones that can easily be cultured at room temperature[26–28]. They also tolerate freezing[20]: this species can be stored at −80 °C for several years prior to hatching[28].

Here, we present a chromosome-level assembly of the 326 million base pair (megabases, Mb) *Ephydatia muelleri* genome. The highly contiguous assembly of this sponge genome is an exceptionally rich resource that reveals metazoan typical regulatory elements, macrosynteny shared with other non-bilaterians and chordates, and allows analysis of structural chromatin variation across animals. The high gene count of sponges compared to other animals is shown to be a feature of gene duplication. These analyses, together with the evaluation of RNA expression in development and host–microbe relationships across the range of this species provide key data for understanding the genomic biology of sponges and the early evolution of animals.

## Results and discussion

**A chromosomal-level genome, with higher gene content than most animals.** We have produced a high-quality assembly of the 326 Mb *Ephydatia muelleri* genome using PacBio, Chicago, and Dovetail Hi-C libraries sequenced to approximately 1490 times total coverage (Supplementary Note 2). The resulting assembly has 1444 scaffolds with a scaffold N50 of 9.8 Mb (Fig. 2a–c, Supplementary Note 3) and 83.7% of the genome (270 Mb) is encompassed in the largest 24 scaffolds in the assembly. These 24 scaffolds encompass 22 of the 23 *Ephydatia muelleri* chromosomes ($2n = 46$)[29] as single sequences, with one chromosome represented by two sequences likely split at the centromere (Fig. 2b).

One scaffold (the 25th largest), containing the partial genome of a member of a *Flavobacterium* spp. (3.09 Mb with 3811 genes), a possible symbiont species, was removed from the final assembly and analyzed separately (Supplementary Note 4, Supplementary Data 1. The remaining 1419 scaffolds (53.3 Mb) contain less than 16% of the genome and have no other clear bacterial sequence content (section 3.5 of Supplementary Note 3).

The *E. muelleri* assembly size corresponds well to the predicted genome size based on Feulgen image analysis densitometry (0.34 pg) and flow cytometry (0.33 pg)[30], and is twice the size of the well-studied *A. queenslandica* genome (166 Mb; Supplementary Table 2)[4], but similar in size to *Sycon ciliatum* (genome size 357 Mb)[31]. The genome is approximately 43% G+C (Supplementary Table 2: cf. *A. queenslandica*: 35.82%, *S. ciliatum*: 46.99%). Nearly 47% of the genome is repetitive sequences, compared to *A. queenslandica* 43% and *S. ciliatum* 28% (Supplementary Table 6). The *E. muelleri* genome browser and other resources are available at https://spaces.facsci.ualberta.ca/ephybase/.

Sponges, perhaps counterintuitively, have more genes than most other animals, and almost twice the number of genes found in humans. The *E. muelleri* genome contains 39,245 predicted protein-coding loci (Supplementary Table 10). This number of genes compares well with recent estimates for *A. queenslandica* (40,122)[5], *S. ciliatum* (32,309)[31], and *Tethya wilhelma* (37,416)[8]. The gene annotations contain 90.10% of the 303 eukaryotic

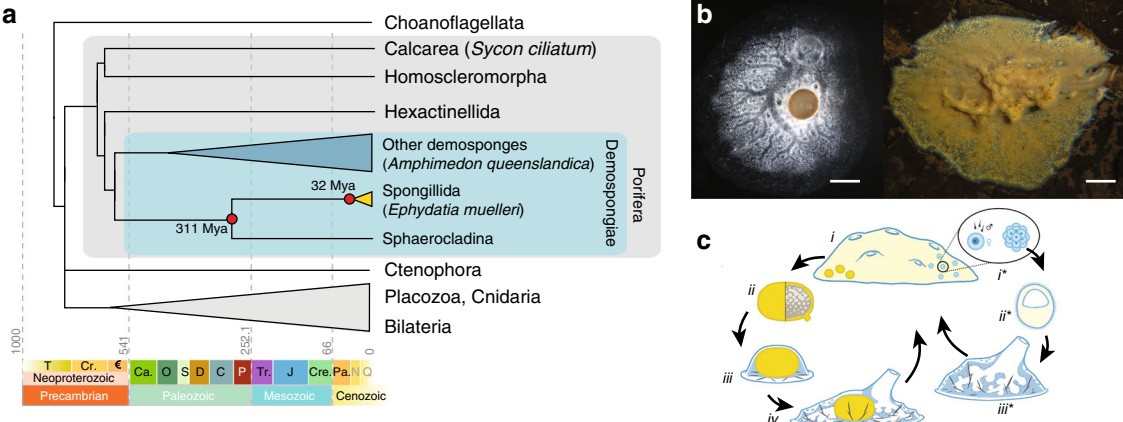

**Fig. 1 *Ephydatia muelleri* biology and inter-relationships. a** Diagrammatic cladogram of poriferan inter-relationships, showing fossil-calibrated divergence clock at bottom and at key nodes. These estimates of divergence were taken from prior analyses as shown in Supplementary Fig. 1. Tree is rooted with choanoflagellate species. Key clades and previously sequenced species are boxed and indicated. **b** Light microscope images of *E. muelleri* in culture and in situ (photos: S. Leys, scale 1 mm in left image, 1 cm at right). **c** Life cycle of *E. muelleri*, showing asexual (left) and sexual (right) modes of reproduction. Gemmules indicated as yellow dots in adult tissue. Asexual reproduction proceeds via hardy gemmules produced internally (i), which are separated from adults (ii), hatch and directly develop (iii) into adult tissue (iv). Sexual reproduction proceeds via generation of gametes (i*), embryonic development into a mobile parenchymella larva (ii*), and development as an independent adult (iii*).

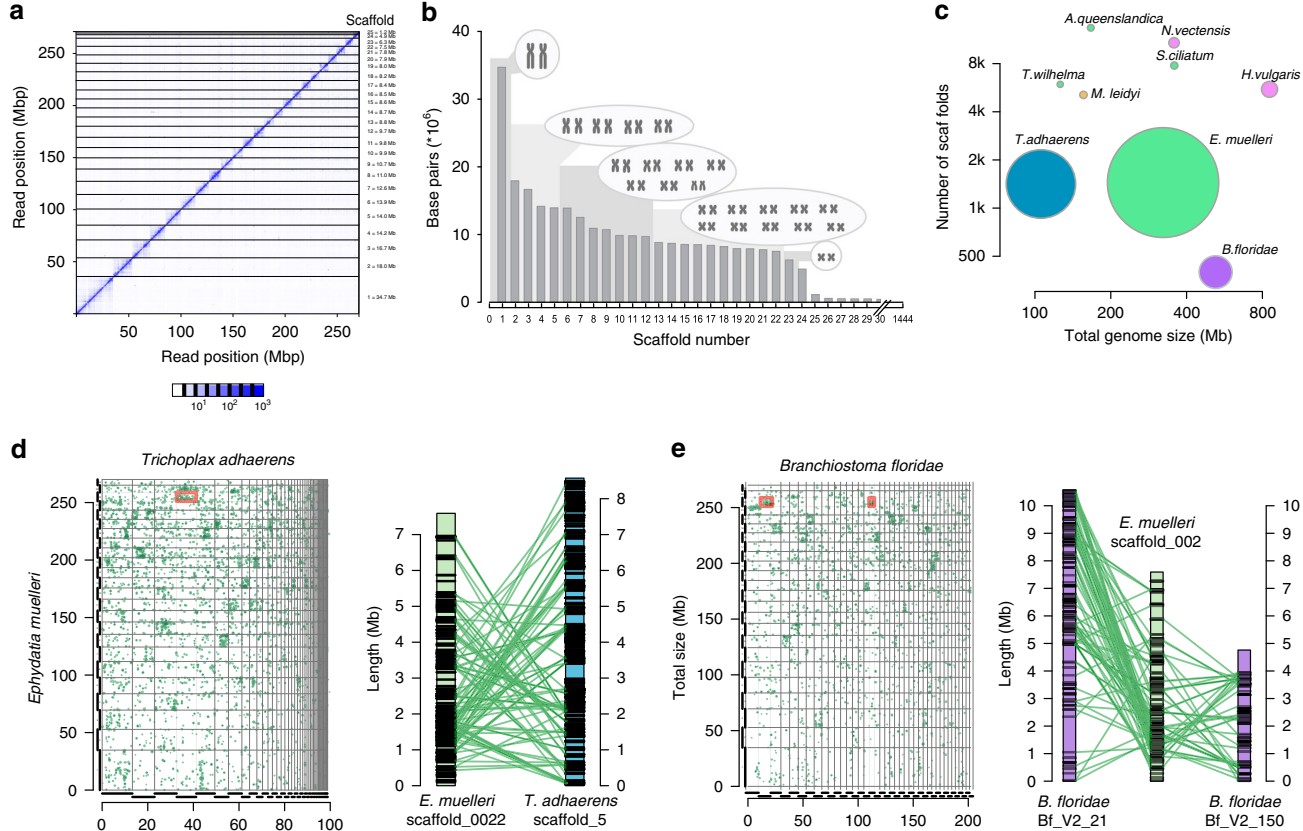

**Fig. 2 Genome assembly and architecture. a** Hi-C contact map, visualised in Juicebox.js, with number of contacts coloured in blue according to the scale below. **b** Histogram of assembly scaffold sizes and corresponding chromosomes drawn to scale after Ishijima et al.[29] Note the large difference in size between chromosome 1 and the additional 22 chromosomes, which was reflected in our assembly. **c** Representation of the relative completeness of the *E. muelleri* assembly compared to those of a number of commonly used genomes. N50 is represented by diameter of circles and number of scaffolds / genome size are on the *x* and *y* axes, respectively. The high level of contiguity of our assembly is obvious in this comparison. **d** Conserved syntenic signal in *Ephydatia muelleri* with *Trichoplax adhaerens*, **e** and with *Branchiostoma floridae*. Each small green dot on the matrices represents an identified homologous gene pair. Individual scaffolds for each species pair possess numerous homologous genes. Red-boxed locations are shown in detail to the right of each matrix as chromosomal representations. Each black line on the chromosome images represents a gene linked in green to the homologous gene in the other species. It is clear that numerous genes are conserved between ancestrally homologous chromosomal regions between these species, although their relative arrangement within chromosomes is shuffled, as expected under the DCJ-DS model of syntenic evolution. Source data are provided as a Source Data file.

BUSCO genes (83.83%, [254 genes] as complete models, section 3.2 of Supplementary Note 3). Approximately 74% of the *E. muelleri* proteins (29,571) have similarity to other organisms as determined by DIAMOND BLASTx (nr database, *E*-value threshold of $10^{-5}$), and nearly half of these hits (13,579) were best-matches to other sponge sequences (section 5.2 of Supplementary Note 5, Supplementary Data 2). We assigned 19,362 genes (approximately 50% of the total number) a full functional annotation using BLAST2GO (section 5.3 of Supplementary Note 5). Intron size and intergenic distance scale with genome size[32] and accordingly both intron size and intergenic distance are greater in *E. muelleri* compared to *A. queenslandica* (Supplementary Table 10, Supplementary Data 3), but these values are still relatively small when compared to other animal genomes.

The abundance of genes seen in *E. muelleri* is in part due to tandem duplication. Many gene clusters have identical intron−exon structure between duplicated genes, suggesting that the mechanism of gene duplication is from replication slippage and unequal crossing-over. *E. muelleri* also shows evidence of widespread segmental duplication, with many gene clusters replicated. For example, both scaffold_0002 and scaffold_0004 contain a large cluster of predicted homologues of integrins, while on scaffold_0004 the integrin cluster overlaps with a large cluster of 177 predicted E3-ubiquitin ligases (e.g., Supplementary Fig. 6). These cluster duplications are recognisable by their close similarity in sequence, especially in coding sequences, but intergenic and intronic sequences are highly variable, strongly suggesting that these clusters are true duplications and not assembly artifacts. Even BUSCO genes, which are found in single copy in most genomes, are duplicated in *E. muelleri* with 19.6% represented by more than one copy (section 3.2 of Supplementary Note 3, Supplementary Tables 4, 5).

**Conservation of synteny with other metazoans.** Sponges diverged from other metazoans in the Neoproterozoic (540–1000 mya) and yet we found evidence for conserved syntenic regions and even local gene order within scaffolds between *E. muelleri* and *Trichoplax adhaerens*, *Nematostella vectensis* and the chordate *Branchiostoma floridae*. Synteny conserved over hundreds of millions of years is consistent with the hypothesis that gene shuffling primarily occurs within chromosomes rather than between them, as predicted by the double cut and join-dosage sensitivity (DCJ-DS) model[33]. The DCJ-DS model predicts that dosage sensitive genes would tend to stay on the same chromosome, although the local order may change. We find this is clearly observable in shorter chromosomes (Fig. 2d). For example, scaffold_0022 in *E. muelleri* matches with scaffold_5 in *T. adhaerens*, scaffold_3 and scaffold_16 in *N. vectensis*, and scaffolds Bf_V2_21 and Bf_V2_150 in *B. floridae* (Fig. 2d, e). Only ten proteins from scaffold_0022 are shared by all species. Two are SLC36/VIAAT-group transporters (Em0022g323a and Em0022g324a), one is a predicted homologue of the mitochondrial enzyme ETFDH (Em0022g346a), and another is a predicted homologue of the splice factor A-kinase anchor protein 17A (Em0022g347a). However, 68 proteins from scaffold_0022 are found in 2 out of 3 species, generally in both *T. adhaerens* and *B. floridae*. None of these species display co-linearity with *E. muelleri* in either region, indicating that while the genes match, sequential order is lost, a typical hallmark of macrosynteny. However, overall, these syntenic blocks are comparable and represent ancestrally shared blocks of homologous genes, conserved from *E. muelleri* to *T. adhaerens* and *B. floridae*, and thus can be inferred to represent ancient groupings conserved since the common ancestor of sponges and bilaterians. While macrosynteny has been found across genomes with high levels of

contiguity[33], previous sponge genomes did not suggest this pattern, largely due to their comparatively fragmented assemblies[4] (Fig. 2b).

In contrast to the conserved synteny between animal lineages, we detected no conserved syntenic regions between *E. muelleri* and two choanoflagellates, *Monosiga brevicollis* and *Salpingoeca rosetta* (Fig. 2d, e and section 3.6 in Supplementary Note 3, Supplementary Fig. 8, Supplementary Data 4). While *E. muelleri* does not show clear macrosyntenic conservation with either of the two choanoflagellates examined here, the two choanoflagellate species do show shared synteny with each other (Supplementary Fig. 8). This disparity between the choanoflagellate and metazoan gene orders suggests that gene macrosynteny has been shuffled either in the lineage leading to the ancestor of choanoflagellates (*Monosiga* and *Salpingoeca*), or in the lineage leading to the last common ancestor of sponge and other animals, or, alternatively, in both of these lineages.

**Pan-metazoan epigenetics.** The large number of genomes now available make it clear that differences in gene regulation, as well as gene content, are responsible for the innovations seen in different animal body plans and phyla[34]. To understand the processes underlying gene regulation in non-bilaterian metazoans, data on the three-dimensional architecture of chromatin is needed from these clades. We used HOMER and Bowtie2 to analyse Hi-C data from *E. muelleri* and found that as in other animals the genome is organised into topologically associating domains (TADs) as well as loops, although we did not find mammalian-like corner peaks at the edges of the predicted TADs (Fig. 3a). These TADs are slightly larger on average than those seen in *Drosophila melanogaster*, at 142.4 kbp, compared with ~107kbp[35]. As in other non-bilaterians, the *E. muelleri* genome lacks the CCCTC-binding factor (CTCF) but does possess a suite of non-CTCF zinc finger proteins which form a sister group to the bilaterian CTCF proteins[36]. Besides these CTCF-like sequences, the *E. muelleri* genome contains both cohesin and structural maintenance of chromosome (SMC) sequences, which are highly conserved throughout eukaryotes (section 6.1 in Supplementary Note 6).

Cytosine DNA methylation is an important part of genome regulation in animals, where transcriptionally active gene bodies are methylated. However, it has recently been shown that, like vertebrates and unlike most invertebrates, sponges have highly methylated genomes in both gene bodies and intergenic regions[9]. Nevertheless, that analysis only sampled a single demosponge (*A. queenslandica*) and a single calcareous sponge (*S. ciliatum*), which suggested sponges may have highly variable levels of genomic methylation. To assess whether the high levels of methylation seen in *A. queenslandica* are common to other demosponges, we carried out whole genome bisulfite sequencing on tissue from a fully developed (Stage 5) *E. muelleri* genomic DNA sample. The global genomic level of methylation in *E. muelleri* is 37% (mCG/CG), which is higher than most invertebrates profiled to date[9], but much lower than *A. queenslandica* (81%) and *S. ciliatum* (51%) (Fig. 3b). The slightly higher repeat content of *E. muelleri* compared to *A. queenslandica* (47 and 43% respectively, Supplementary Table 6) indicates that hypermethylation in *A. queenslandica* cannot be driven by an exceptionally high repeat content in that species. The *E. muelleri* methylome thus challenges the assumption that all demosponges have hypermethylated genomes, and suggests that the *A. queenslandica* pattern is a lineage-specific innovation. Whether methylation levels differ significantly in freshwater compared to marine environments has yet to be explored, especially in invertebrate taxa, and could have a bearing on this inference.

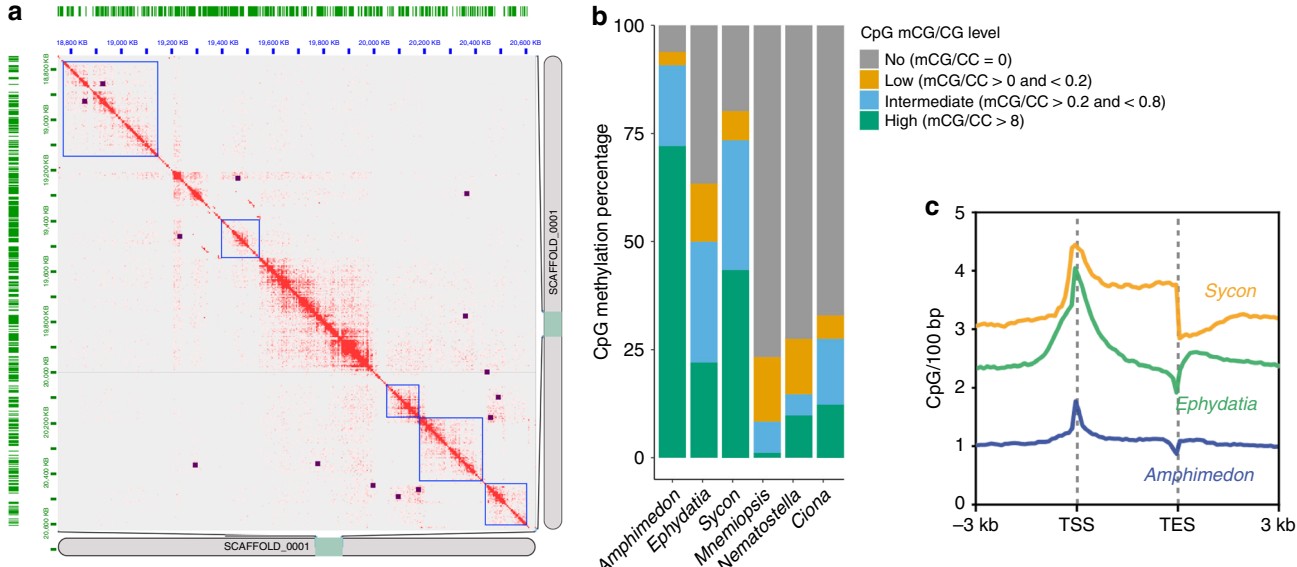

**Fig. 3 Gene regulation. a** Hi-C contacts within scaffold (=chromosome) 1 of our assembly, showing contacts in red. Also shown are loops (purple dots) and topologically associating domains (TADs, blue squares) as assessed by HOMER. Note that we do not see mammalian-like corner peaks at the edges of TADs (which would appear as loops at the corner of TADs). **b** Presence and absence of methylation at CpG sites in 3 species of sponge and 3 previously studied metazoan species. Note that *Amphimedon queenslandica* and *Sycon ciliatum* are highly methylated, while *Ephydatia muelleri* is more modestly methylated, although not quite at the levels seen in *Mnemiopsis leidyi*, *Nematostella vectensis*, or *Ciona intestinalis*. Also shown is the arrangement of possible methylation sites (CpG) relative to transcriptional start sites (TSS) and transcriptional end sites (TES) in sponge genomes. In *A. queenslandica* these are only slightly enriched at TSS, while in both *S. ciliatum* and *E. muelleri* these are highly enriched (CpG/100 bp) near TSS. *S. ciliatum* maintains these levels across the transcribed region, while they decline in abundance in *E. muelleri*. In all species TES show a slight depreciation of CpG site abundance relative to other regions of the genome. Source data are provided as a Source Data file.

Since cytosine methylation is highly mutagenic, vertebrate and *A. queenslandica* genomes are highly depleted for CpG dinucleotides[9]. Congruent with the intermediate methylation levels, we found that the genome of *E. muelleri* is also depleted for CpG dinucleotides, more than most invertebrates but less than in *A. queenslandica* (Supplementary Fig. 19). However, CpG content varies greatly across sponge genomes; for instance, *S. ciliatum* has higher methylation than *E. muelleri*, but has a relatively higher amount of CpGs. This indicates that CpG depletion is not fully coupled to methylation levels in sponges, and that retention of CpGs might obey unknown species-specific constraints.

Given that *E. muelleri* shows methylation levels more consistent with canonical mosaic invertebrate methylomes than with a hypermethylated genome, we then checked whether gene body methylation accumulation is dependent on gene transcription. CpGs are more commonly observed near transcriptional start sites (TSS) than in *A. queenslandica*, but marginally lower in absolute levels than those seen in *S. ciliatum* (Fig. 3c). As observed in many invertebrates, *E. muelleri* genes with mid-transcriptional levels show higher gene body methylation than non-expressed genes or highly expressed genes (Supplementary Fig. 19B)[37]. Promoters are strongly demethylated and repeats found within gene bodies tend to have higher methylation levels than those in intergenic regions, as seen in other invertebrates[38], suggesting that not all repeats are actively targeted by DNA methylation in *E. muelleri*. In fact, repeat methylation level positively correlates with age of the repeat, and LTR retro-transposons are more likely targeted by DNA methylation irrespective of genome position (Supplementary Fig. 19). Overall, the *E. muelleri* methylome shows many patterns similar to those of canonical mosaic invertebrate genomes, and may therefore provide a more appropriate comparison for future comparative epigenetics work than other existing sponge models.

**Sponges show high levels of gene gain**. Every sponge species we examined showed a gain of 12,000 more genes since their divergence from the most recent sister taxon or clade (Fig. 4a, section 7.1 in Supplementary Note 7, Supplementary Fig. 20). The large number of duplicates we identified in *E. muelleri* and other sponges is consistent with recent independent findings[34], and can be traced to the lineage leading to the divergence of the freshwater order Spongillida from the marine Heteroscleromorpha (Supplementary Fig. 21). This finding is robust to different placements of sponges relative to other metazoan taxa (Fig. 4a, b) and further suggests a role for duplication and gene gain in freshwater evolution[34].

Despite a high rate of gene gain, we observed no greater number of losses in the freshwater lineage than in other groups, and an equal number of lineage specific expansions for all taxa examined. For example, while sponges have lost 375 orthogroups compared to the last common ancestor of animals, 1340 are inferred to have been lost in the ctenophore lineage assuming that Porifera are the sister taxon to other Metazoa, or 1812 if ctenophores are assumed to be sister to other Metazoa (Fig. 4a, b, section 7.1 in Supplementary Note 7, Supplementary Fig. 21). The large numbers of genes found in sponges can therefore largely be explained by steady rates of gain in genes via duplications that are not matched by similarly high rates of gene loss.

**Molecular signals of freshwater adaptation**. To determine whether transitions to freshwater are accompanied by the loss of a common set of genes in independent clades, we studied shared losses in four disparate animal lineages, using pairs of species for each lineage, in which one is marine and the other is freshwater. Gene gain is also noted in section 7.4 in Supplementary Note 7, Supplementary Fig. 23, although these gains are lineage specific. Our dataset consisted of: Porifera: *A. queenslandica* and

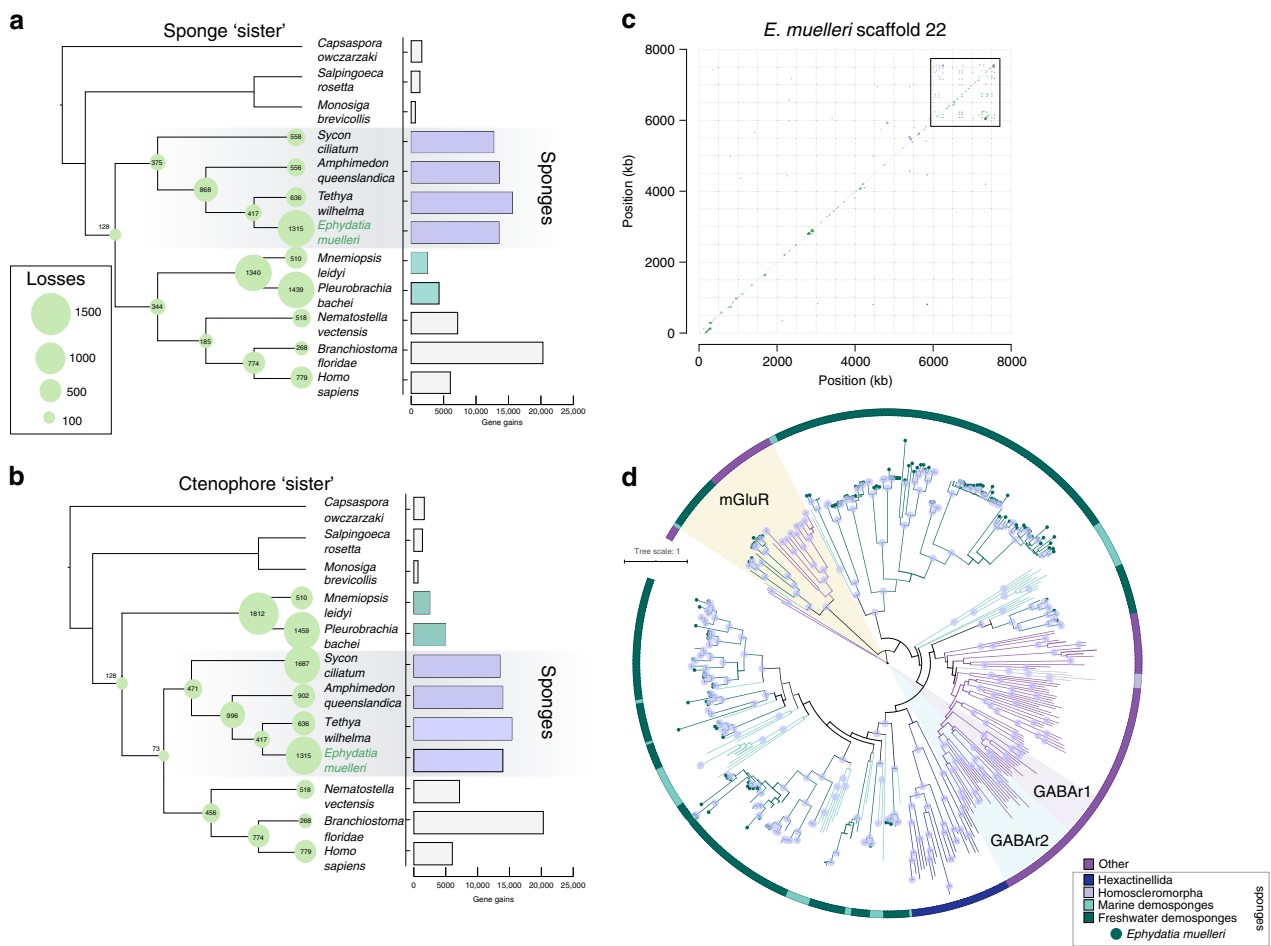

**Fig. 4 Gene gain and loss and its role in freshwater evolution.** Gene gains (histogram at right) and losses (numbers in circles at nodes) across sponge and metazoan phylogeny, as assessed using a selection of choanoflagellate, sponge, ctenophore, and eumetazoan species with full genome sequence available, assuming sponges (**a**) or ctenophores (**b**) are sister to other metazoans, as shown on representative cladograms. Note that the large number of apparent gains in *Branchiostoma floridae* is largely due to incompletely curated records in that resource. Sponges show a uniformly high rate of gene gain relative to other metazoans. *E. muelleri* shows considerable gene losses, but this is likely less acute in reality due to incomplete gene models. **c** Example of a highly duplicated gene in *E. muelleri*, the m*GABA* receptor. Here we show the incidence of this gene, as a segmental duplicate dotplot, on scaffold 22. Same strand (forward) matches are represented in green, and reverse strand matches in purple. Boxed is the cluster of extremely high duplication levels of this gene. **d** Phylogeny of *mGluR* and *mGABAR* genes, rooted with *Capsaspora owczarzaki* sequences. Phylogeny generated in IQTREE v1.6.9 under the WAG + F + R9 model, based on a 1364-position (appx 45% gaps) amino acid alignment generated in MAFFT v7.313 (with -linsi options). Tree visualised in iTOL, with *mGluR*, vertebrate *GABAR1* and *GABAR2* genes shaded. Dots on nodes represent nodes with 100% bootstrap support. Colour of branches and outer circle indicates origin of sequences, with *Ephydatia muelleri* and other species coloured as seen in Legend, bottom right. Note the extreme level of duplication of these genes seen in sponges in general, and in freshwater lineages in particular. Source data are provided as a Source Data file.

*E. muelleri*; Cnidaria: *Nematostella vectensis* and *Hydra vulgaris*; Annelida: *Capitella teleta* and *Helobdella robusta*; and Mollusca: *Lottia gigantea* and *Dreissena polymorpha*. We found that, of the more than 30,000 orthogroups within which we identified losses specific to freshwater species, there were 29 shared losses in all four freshwater lineages compared to 4 shared losses in marine species (section 7.4 in Supplementary Note 7, Supplementary Data 5), while the average loss rate for any 4 taxa across our sample was 16.9 genes. We also found significantly higher ($p = 0.013$, *t*-test) numbers of shared orthogroup losses in three of four freshwater lineages, compared with the direct marine counter-examples of such a pattern (73, 22, 52, and 37, cf. 1, 2, 8, and 0 same-phylum marine losses, section 7.4 in Supplementary Note 7, Supplementary Data 5). Shared loss in freshwater lineages therefore seems to be an infrequent phenomenon, whereas it happens rarely across marine taxa.

The transition to fresh water, and the more recent radiation of extant species, has left signatures of positive selection in the Spongillidae[24], and in *E. muelleri* in particular. Using multiple tests, we found 117 orthogroups to be under positive selection in *E. muelleri* alone, 23 of which were also under positive selection in all freshwater sponges (section 7.3 in Supplementary Note 7, Supplementary Data 6). The move to freshwater conditions must be accompanied by a diverse range of changes to membrane functionality. Several genes known to perform roles in homeostasis and membrane function including *V-type proton ATPase subunit B*, three kinds of *sorting nexin*, *vacuolar-sorting protein SNF8*, and *Multidrug and toxin extrusion protein 1*, were found to be under particularly high selection pressure (Supplementary Table 11). Almost all of the *E. muelleri* genes in these orthogroups, 85 of 117, are differentially expressed across the process of development, underlining their importance to *E. muelleri* biology (Supplementary Fig. 22E). It is not uncommon for these differentially expressed genes to have multiple in-paralogs. The most prolifically duplicated genes are a *cytoplasmic actin* and a *leukotriene receptor* which are both tandemly

duplicated 6 times in the *E. muelleri* genome, from loci Em0009g1201 and Em0009g943 respectively. Altogether, of the 85 differentially expressed, positively selected orthogroups, 54 are single copy, and 31 possess two or more in-paralogs of the genes tested (Supplementary Data 6). This indicates that duplication of these genes is commonly associated with adaptation.

Gene duplication is known to be a means of adaptation to new environments more generally[39], with sub-functionalisation and neo-functionalisation allowing specific changes to molecular function in response to changing conditions. We explored genes that have expanded in number in the *E. muelleri* genome compared to other sponges, ctenophores and chordates (here represented by *B. floridae* and humans). We found that the largest clusters included genes involved in chemokine binding, and one cluster included over 50 *metabotropic GABA receptors* (section 7.5.1 in Supplementary Note 7). More than 120 *mGABA* receptors are predicted in total in the *E. muelleri* genome (compared to two in humans), 48% of which are on scaffolds 4, 13 and 22. Most of these are expressed, alongside other enlarged gene complements such as *cortactin* (52 clusters), *NBAS* (35) and *integrin beta* (36) (Supplementary Data 7).

Another gene showing high levels of duplication in *E. muelleri* is *aquaporin*, a water and solute carrier known to be involved in freshwater adaptation[40]. Demosponges possess AQP8 (a family of aquaporins that allow the passage of mostly water, but also ammonia and urea), as well as aquaglyceroporins (AQP3,7,9 and 10), which control movement of glycerol, arsenite, and silicic acid among other compounds. However, in freshwater sponges only aquaglyceroporins are present, while many aquaporins are lost compared to outgroups (Supplementary Fig. 28). Freshwater sponges, like *Hydra* and many protists, use contractile vacuoles to excrete water[19] and so it is possible that the duplication of aquaglyceroporins in freshwater sponges may have allowed some of the genes to take on new functions. For example, in mammals, AQP9 can mediate silicon influx in addition to being permeable to glycerol and urea, but not to water itself[41]. Since sponge aquaglyceroporins are more similar to AQP9 than to AQP3 or 7 (section 7.5.2 in Supplementary Note 7 and Supplementary Fig. 28) it is possible that in freshwater sponges, in particular, these gene families function in silicon transport for skeletogenesis. Aquaporin-like molecules, glycerol uptake facilitator proteins (GLFPs), are found in bacteria and plants but, to date, have not been detected in animal genomes[42]. *E. muelleri* has nine paralogs of GLFP, with five of them located on the same scaffold (Em0019) (Supplementary Fig. 28). We hypothesise that, as in plants[42], the presence of GLFPs in sponges came about via horizontal gene transfer.

**Gene expression during *E. muelleri* development**. To understand what genes are common and which are distinct from other metazoans during the development of the filter-feeding body plan, we examined differential gene expression from hatching gemmules through to the formation of a filtering sponge. The majority (32,690/39,245) of the *E. muelleri* gene complement was expressed at some point in the course of development. Remarkably, over 33% of the total gene models were differentially expressed (log2 (fold change) >1/<−1) across the gemmule-hatching process (Fig. 5, Supplementary Note 8). The pattern of gene expression shows a typical shift that occurs during the development of animals from the breakdown of reserves stored in cells in the embryo (here thesocyte stem cells stored in overwintering gemmules) at Stage 1. Stages 2 and 3 show activation of developmental patterning genes, genes involved in cell motility and the production of extracellular matrix. The upregulation of genes involved in structural maintenance, homoeostasis, and the

immune system occurs at Stage 5 (Fig. 5 and Supplementary Note 8). At Stage 1, arachidonate pathways for glycogen breakdown and fatty acid metabolism were differentially upregulated to produce the breakdown of stored reserves (Fig. 5 and Supplementary Note 8). Many of the genes involved in formation of a basement membrane and true epithelia were originally considered to be eumetazoan[43], but we found in the *E. muelleri* genome, genes for *type IV collagen, contactin, laminin, PAR3/6, patj, perlecan* and *nidogen* that exhibit gene expression profiles consistent with their known role in development of polarised epithelia (Fig. 5 and Supplementary Fig. 31). Similarly, *claudin*, which may be involved in the tight seal that *E. muelleri* epithelia have been shown to form[44], is expressed later in developmental time. While eukaryotic genes are expressed throughout the different developmental stages, many sponge-specific genes are expressed only as the sponge hatches and develops the aquiferous system that is common only to Porifera. Expression profiles of Wnt, TGF-β and Hedgehog signalling pathways are given in Supplementary Fig. 36 and indicate that the greatest difference in transcription occurs between hatching and the early development of the sponge specific characteristics of spicules, chambers and the aquiferous system (details, section 9.4 in Supplementary Note 9). Individual components of these pathways are expressed at discrete time points in the process of development. For example, in the Wnt pathway, three of the secreted frizzled receptor proteins (SFRP B, SFRP E, SFRP F) as well as two LRP receptors (LRP 2, LRP 4B) are upregulated during hatching from the stem cells stored in the gemmule (Stage 1), and in the TGF-β pathway, receptors are expressed in Stages 1–3, but downregulated thereafter. This data therefore provides a wealth of information for understanding and contrasting the genetic processes underpinning sponge development.

**Sensation and non-nervous signalling in sponges**. Sponges have no nervous system and yet they contract in response to a range of stimuli[45]. Exactly how contractions are coordinated is still unknown, and the potential position of ctenophores rather than sponges as sister to the rest of animals on the tree of life[46–48] provocatively implies that sponges could have lost neurons.

It has been difficult to identify a single character of neurons shared by all animals[49], but the synapse and in particular the proteins that compose its scaffolding and chemical neurotransmitter complement, are agreed to be an important component[50]. One difficulty is that genes in the neuronal synapse also have other tissue-specific functions. One family of genes with tissue-specific functions are the SyNaptosomal-associated proteins (SNAPs) of vertebrates. These proteins are members of the Soluble N-ethylmaleimide sensitive factor attachment protein REceptor (SNARE) protein superfamily[51] which function in membrane fusion at the cell surface. In mammalian cells SNAP-25 mediates the fusion of vesicles with the presynaptic membrane of neurons, while its paralogue SNAP-23 mediates the fusion of vesicles in other regions of the cell surface[52]. The presence of genes encoding SNAP-23/25-like SNARE proteins has sometimes been inferred to indicate presence of neuron-specific protein machinery[53]. However, homologues of SNAP SNAREs are widely conserved among eukaryotes without nervous systems, such as plants[54]. This raises the question of whether particular SNAP SNARE genes found in early-branching metazoan lineages indicate an early origin of neuron-specific protein machinery.

Our phylogenetic analysis of SNAP-23/25 homologues revealed that the vertebrate neuron-specific paralogue SNAP-25 arose from a duplication that occurred in the vertebrate stem lineage, while the two SNAP-23/25-like genes found in *E. muelleri* arose from an independent duplication that occurred in Porifera

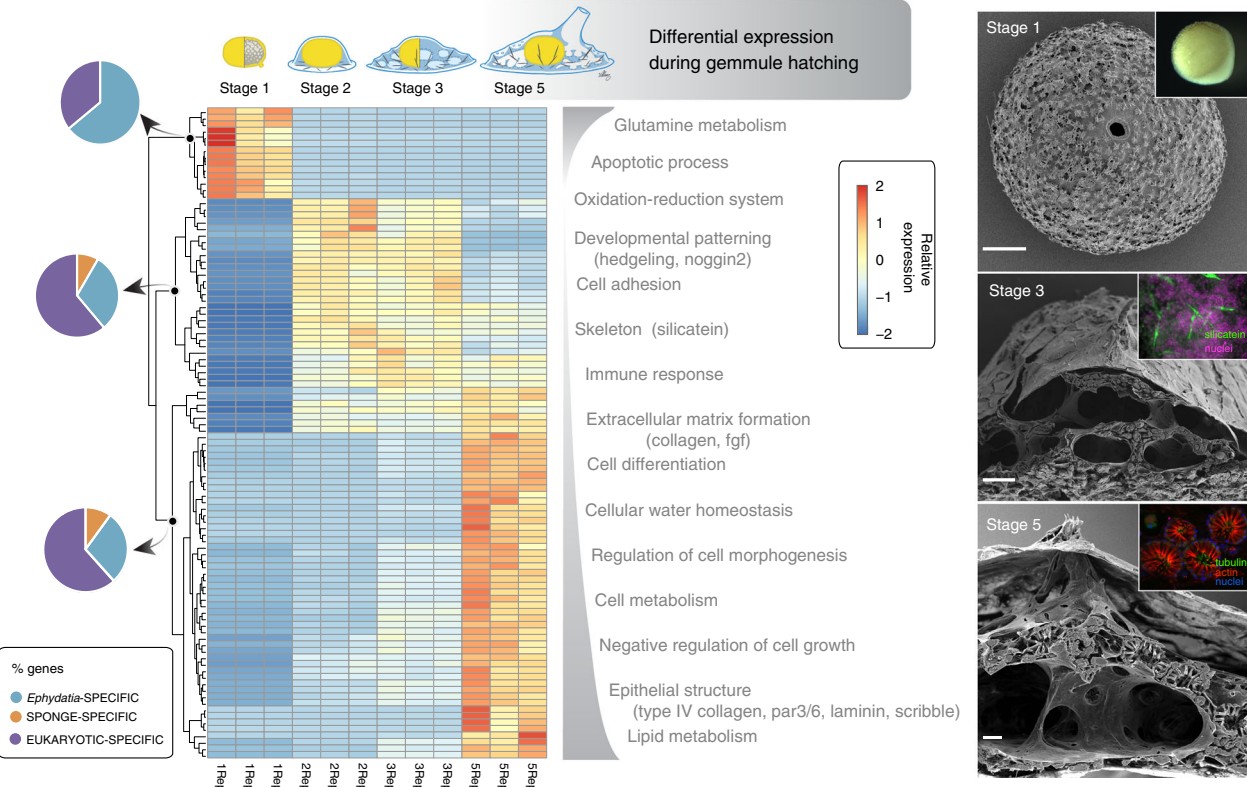

**Fig. 5 Gene expression across *Ephydatia muelleri* development.** Heatmap shows relative expression of differentially expressed genes (scale at right of heatmap) across the course of *E. muelleri* development, with triplicate samples for Stages 1–3 and 5 of growth. Note the large changes in gene expression that occur between Stages 1 and 2, and between Stages 3 and 5. These changes in expression are coincident with the occurrence of a silica-based skeleton (beginning Stage 2) and a fully formed aquiferous system including an osculum (Stage 5). Also shown on this figure are the GO terms matched by these genes (categories at right of heatmap) and the origin point of these genes—whether they are sponge/*Ephydatia* novelties, or found more generally across the animal tree of life. *E. muelleri* specific genes are found more commonly in Stage 1, likely a result of the specialist needs of the gemmule stage tissue represented by this sample. Images at right are of (top: bottom) a Stage 1 gemmule, taken with an SEM/light microscope (inset), a Stage 3 hatching sponge, showing the occurrence of spicules (see image top right, labelled with silicatein in green), and a Stage 5 juvenile sponge, with a well-developed aquiferous system (inset shows choanocyte chambers with collars labelled red for actin, flagella green for tubulin and nuclei, blue). Scale bars: top 100 μm, middle 50 μm, and bottom 10 μm. In these images only one replicate is shown for illustrative purposes, but in all cases the number of individual gemmules or sponges observed with these patterns was >10, for >3 replicates. Source data are provided as a Source Data file.

(Supplementary Fig. 29, Supplementary Data 8). This means that, like their non-holozoan homologues, none of the identified poriferan SNAP-23/25-like genes are more closely related to SNAP-25 than to the non-neuronal vertebrate paralogue SNAP-23. Given the high quality of the assembly of the *E. muelleri* genome, this result shows that SNAP-25 synapses arose after sponges diverged from the rest of animals, and this is consistent with a late origin of synaptic type electro-chemical signalling in the metazoan stem lineage, after the divergence of Porifera.

One overt behaviour of *E. muelleri* is a series of convulsions, which it uses to dislodge particles clogging its collar filters[26]. Previous work indicated that sensory cilia in the osculum were required for effective contractions and implicated a role for transient receptor potential (TRP) channels in sensing changes in water moving through the sponge[55]. We found a large diversification of TRP channels in the *E. muelleri* genome, and these grouped with the TRPA and TRPML families (section 9.1 in Supplementary Note 9). There is differential loss of TRPM, TRPML, TRPVnan, TRPV and TRPP2 in each of the four major lineages of sponges, but sponges as a group have lost TRPC/TRPN channels, as homologues of that group are known from choanoflagellates (Supplementary Note 9). TRPA genes are some of the best characterised and are known as mechano- or chemo-receptors whereas TRPML families are largely considered to be

expressed on organelles inside cells[56]. The diversification of TRPA channels in *E. muelleri* and other demosponges suggests a molecular mechanism for mechanoreception as well as chemical sensation in this clade.

In the *E. muelleri* genome we also found a wide range of ion channels involved in signalling in eumetazoans (Supplementary Note 9), but there are conspicuous absences including voltage-gated sodium and potassium channels, epithelial sodium-activated channels (ENaCs), leak channels, and glutamate-gated ion channels (GICs). Also absent are receptors for monoamine (serotonin and dopamine) signalling, as well as key components of the biosynthesis pathways for these, as well as ionotropic glutamate receptors. While the latter are present in calcareous and homoscleromorph sponges and in non-metazoans, demosponges seem to have lost them. In contrast, we found evidence for a diversity of metabotropic glutamate receptors (mGluR), as well as a wealth of metabotropic GABA-receptors, as discussed above.

In *E. muelleri* therefore, as in other demosponges, there is evidence for components that allow sensation of the environment via TRP channels, among others, and non-neuronal chemical signalling via metabotropic GPCRs (e.g., receptors for glutamate and potentially GABA and/or a range of organic acids), but no evidence for more rapid electro-chemical signalling. While we find no signature for any aspect of conventional nervous tissues in

the *E. muelleri* genome, we cannot rule out the possibility that the phylum Porifera as a whole, or individual lineages within it (including *E. muelleri*), have lost these neuron-related components.

**Host–microbe associations in *E. muelleri*.** Most animals possess diverse symbiotic microbial consortia, which provide their hosts with metabolic advantages and new functions, and sponges are no different[57,58]. The release of the genome of *A. queenslandica*[4] opened a window into the study of the mechanisms of sponge–microorganism interactions[57]. To unravel the recognition mechanisms developed by host and microbes to facilitate symbioses, high quality genomes (and more genomic resources in general) are fundamental.

The genome of *E. muelleri* offers a model that allows exploration of eukaryotic patterns of microbial recognition in unique environments. We studied the microbiome of 11 different specimens of *E. muelleri* collected from six locations across 6500 km in the Northern hemisphere, and found that this species contained between 865 and 4172 unique amplicon sequence variants (ASVs) (Supplementary Note 10, Supplementary Data 9). The microbiome of *E. muelleri* has a level of diversity comparable to that of the most diverse marine demosponges[58,59]. The microbiome of all specimens of *E. muelleri* is largely dominated by Proteobacteria and Bacteroidetes, as in other demosponges (Supplementary Note 10). However, like other freshwater sponges, *E. muelleri* possesses a large fraction of the order Betaproteobacteriales[59], absent in marine sponges, which are traditionally associated with Gammaproteobacteria, a difference which is likely due to the differing pH and nutrients found in the two environments. Surprisingly, even though the entire genome of an unknown species of *Flavobacterium* was recovered during the genome assembly (Supplementary Note 4), Flavobacteriales were not especially abundant in the other *E. muelleri* samples, only reaching 16% relative abundance in adult tissue from UK samples (Supplementary Note 10).

Overall, differences in microbiome content were determined by geographic location, as has been found in marine sponges[58]. For example, only the samples collected from the Sooke Reservoir had a high abundance of Firmicutes and Campylobacteria. Likewise, only those samples collected from Maine had a moderate abundance of Cyanobacteria. Despite the distance separating samples, and therefore potential different ecologies of the collection sites, we found that four ASVs were shared among all samples, yet with different percentages ranging from <1% to >20% in different samples. These four ASVs were assigned to Burkholderiaceae (order Betaproteobacteriales) and Ferruginibacter (order Chitinophagales), and one was an unclassified bacterium (Supplementary Note 10). Whether these ASVs are fundamental for the metabolic function of *E. muelleri*, or whether they are simply cosmopolitan bacteria transported by the wind or on animals, and taken up by all sponges in lakes and rivers, is still to be determined. These findings and resources open the door to studies of species-specific patterns of host–microbe association at a broad scale.

**Conclusions.** The high quality of the *E. muelleri* genome provides a new basis for comparative studies of animal evolution. To date we have lacked a chromosomal-quality poriferan genome assembly, and with this in hand for an experimentally tractable organism, comparative studies of a variety of ancestral characters, including longer-range gene regulation and genomic architecture, become possible.

Given their apparent anatomical simplicity, it can be surprising to some researchers that sponges have nearly twice the gene complement of other animals, but the high quality of this genome confirms that this is not an artefact of previous genome assemblies, and suggests that gene duplication and adaptation to novel environments are responsible for the high gene counts. Sponges possess complex filtering behaviours, integrate with an extensive network of microbes, and have an extensive defence system. As only approximately half of the genes found in sponges can be firmly identified, it is clear that there remains a huge amount of hidden biology yet to be understood in sponges, just as in other non-bilaterians[60]. The robustness of the *E. muelleri* genome and model is an excellent tool for performing this work. It also opens the door to comparative analysis of the genomic changes required for the challenging process of adaptation to freshwater, and to finding out whether these are shared convergently in disparate phyla. Complemented by additional RNAseq, methylation data, and the analysis of symbiont content, the *E. muelleri* genome offers an important new opportunity for exploring the molecular toolkit, from protein coding to gene regulation, that underpinned the early evolution of animals and their diverse, complex, and successful traits.

## Methods

**Sequencing and assembly**. Tissue used for DNA sequencing was derived from a single clone collected as overwintering cysts (gemmules) from the Sooke Reservoir, at the head tank of the city of Victoria, British Columbia drinking water system. A voucher specimen is deposited with the Royal British Columbia Museum (RBCM019-00140-001) (Supplementary Notes 1 and 2). Tissue from a single clone hatched and grown under sterile conditions was flash frozen and stored at −80 °C. DNA isolation and sequencing was carried out by Dovetail Genomics (Scotts Valley, CA, USA) using PacBio sequencing for de novo assembly with Arrow (genomicconsensus package, PacBio tools) followed by preparation of Chicago and HiC libraries that were sequenced on Illumina platforms, and subsequent assembly using HiRiSE (Supplementary Note 2). Genome assembly metrics were determined using a range of tools, and further details of all methods used are available in Supplementary Note 2.

**Genome annotation**. Gene models were predicted using AUGUSTUS 3.3.2 annotation software (http://bioinf.uni-greifswald.de/augustus/) with previously published RNAseq datasets used for training. As the basis for gene prediction, the non-masked genome was used, to avoid artefacts, missed exons or missing gene portions caused by masked areas of the genome. The BUSCO v2/3 set[61] was used in genome mode to determine gene recovery metrics. RepeatModeler 2.0 and RepeatMasker 4.1.0 were used sequentially to predict repetitive content within the genome as described in Supplementary Note 3. Contamination and bacterial content was excluded by BLAST 2.10.0 against a range of well-annotated databases. Syntenic relationships were assessed using reciprocal blasts and custom Python 3.7 scripts (scaffold_synteny.py, see bitbucket repository http://bitbucket.org/ EphydatiaGenome/ or available in Supplementary Data 2). Taxonomy assessment of identified symbiont sequence was performed in MiGA[62] and other software, as detailed in Supplementary Note 4.

Automated annotation of gene sequences was performed using DIAMOND 0.9.31 BLASTx[63] against the *nr* or Swiss-Prot databases followed by functional annotation. Full details are described in Supplementary Note 5. Methylation studies were performed using the MethylC-seq protocol[64] and as described in Supplementary Note 6. TADs and loops were identified using HOMER v4.11[65]. Orthogroup based analyses were performed primarily using Orthofinder2[66], with IQTREE v1.6.12, MAFFT 7.450, and DIAMOND BLAST options (described in detail Supplementary Note 7). Selection tests were performed according to the methodology put forward by Santagata (https://github.com/Santagata/Select_Test) and Kenny et al.[24] and detailed in Supplementary Note 7.3.

**Developmental gene expression**. RNAseq was performed by LC Sciences (Houston, TX). HISAT 2.0[67] was used to map RNAseq reads to the reference *E. muelleri* genome. edgeR v3.14.0[68] was used to estimate the expression levels of all transcripts across all replicate samples. Full details are provided in Supplementary Note 8.

Gene family content was assessed using targeted manual BLAST, with HMMER v3.3[69] used when necessary to test absence. Reciprocal BLAST was used to ensure assignment of identity, with the identity of key gene families assessed using phylogenetic inference as shown in Supplementary Note 9. Holobiont content from a number of *E. muelleri* samples was assessed with Mothur v.1.41.3 and an adaptation of MiSeq SOP protocol[70] as fully described in Supplementary Note 10.

**Reporting summary**. Further information on research design is available in the Nature Research Reporting Summary linked to this article.

## Data availability

A browsable version of the genome of *Ephydatia muelleri*, gene predictions, a masked version of the assembly, and a variety of annotation formats are available from https://spaces.facsci.ualberta.ca/ephybase/. Source data are provided with this paper. Supplementary Data 1–9 contain all appropriate additional data for our analyses. These data are also available from https://doi.org/10.6084/m9.figshare.11847195 and the University of Alberta Education and Research Archive https://doi.org/10.7939/r3-exnc-q910 for ease of download. This project has been deposited at DDBJ/ENA/GenBank under the accession JABACO000000000. The version described in this paper is version JABACO010000000. The sequence of *Flavobacterium* sp. has also been uploaded, with accession number CP051546.1. The raw reads have been uploaded to the NCBI SRA at accession number PRJNA579531/ GEO GSE139500. Source data are provided with this paper.

## Code availability

All scripts used in analysis are available as Supplementary Data 3 and 4 and have also been uploaded to https://spaces.facsci.ualberta.ca/ephybase/ and https://bitbucket.org/EphydatiaGenome/. Source data are provided with this paper.

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

## Acknowledgements

Genome and RNA sequencing for this project was funded by a Natural Sciences and Engineering Research Council (Canada) Discovery Grant (RGPIN-2016-05446) to S.P.L. This work was supported by the EU Horizon 2020 MSCAs ADAPTOMICS (grant agreement: IF750937) to N.J.K. and DeepSym (grant agreement: 796011) to C.D.V. The selection test methods applied in this study were facilitated using best practices and scripts provided as part of a Next Generation Sequencing-based workshop sponsored by the National Science Foundation (Award # 1744877 to S. Santagata). G.W., K.J., R.E.R.V. acknowledge funding from the European Union's Horizon 2020 research and innovation programme under the Marie Skłodowska-Curie grant agreement No 764840 (ITN IGNITE). A.L.H. and S.C. acknowledge funding from the National Science Foundation (Award #1555440). J.F.R. acknowledges funding from the National Science Foundation (Award No 1542597). A.R. acknowledges funding from the EU SponGES project (European Union's Horizon 2020 research and innovation programme, grant agreement: 679849). A.R., G.W., W.R.F. acknowledge funding from VILLUM FONDEN (Grant No. 16518). L.B.-C. acknowledges funding from the DFG, grant number: JE 777/3-1. We thank Karen Evans for sending amplicon sequencing samples, Glen Elliott, Angela Bentley (SEM) and Pamela Windsor-Reid (silicatein FISH) for help in providing images for Fig. 5 and Supplementary Fig. 40, and Isabelle Oliver for help in constructing Supplementary Fig. 35. Ilan Domnich drew the sponge illustrations in Figs. 1 and 5. We thank Dr Ferdinand Marlétaz and Dr Jordi Paps for their comments and guidance with this manuscript, and the members of the Leys and Riesgo labs for their myriad help and support.

## Author contributions

S.P.L. conceived of the project, performed gene analysis, wrote and managed the overall direction of this manuscript. N.J.K., W.R.F. and A.R. performed bioinformatic analyses, wrote the manuscript and aided in project coordination. A.L.H. led work identifying genes, and particularly analysis of Wnt family content, wrote the manuscript and aided in project coordination. G.W. helped write the manuscript and aided in project coordination. S.C. aided with many aspects of gene identification. R.E.R.V. contributed specialist analyses on genome architecture and symbiont content. K.J. provided analyses of unique genes in *E. muelleri* and cluster expansions. A.d.M. and R.L. contributed data and analysis of methylation in *E. muelleri* and other species. J.F.R. contributed to studies of loss, and provided specialist bioinformatic assistance. L.B.C. and L.D.B. provided data and analysis on T.R.P. and SNARE genes. L.G. aided in sample collection, storage and processing. M.R. aided with analysis of Hi-C contact data and inference of genome architecture. C.D.V. performed amplicon sequencing, analysis and curation of results. All authors contributed to and approved the manuscript before submission.

## Competing interests

The authors declare no competing interests.
