## [Peer Review File · Nature Communications]

Reviewers' Comments:

Reviewer #1:

Remarks to the Author:

Kenny et al. have sequenced and assembled the genome of the freshwater sponge *Ephydatia muelleri* using a PacBio long-read sequencing, Chicago library, and Dovetail HiC library. The quality of the assembled genome seems to be at the chromosomal level, much better than available sponge genomes. They performed many analyses including DNA methylation, gene expressions from four developmental stages, and amplicon analysis of associated bacteria. The freshwater sponge genome at the chromosomal level will provide a resource on studying early diverging groups among animals. However, the authors jumped to conclusions without examining the gene expressions on adaptation to freshwater environments. In case of the *Daphnia* genome paper (Colbourne et al. 2010), responsive genes to hypoxia environment have been reported with differential expression data. In addition, several points need clarification.

Detailed points:

Line 1: The title is attractive but there is less discussion about animal origins at the chromosomal level. Rather, they claim that the genome offers a rich resource for future study.

Lines 56-57: The analysis in this manuscript does not offer clarity on the adaptation to extreme temperatures and freshwater. Differential gene expression analysis or comparative genomic analysis between relative species should be provided.

Lines 98-101: It is unclear how the time tree was constructed in Fig. 1. Although they referred to Schuster et al. (2018), it reported the Demospongiae phylogeny and did not show the metazoan phylogeny, including the Calcarea, Homoscleromorpha, Bilateria, and so on. Supplementary Fig. 2 shows the new result of metazoan phylogenetic analysis and likely includes more broad interest result.

Line 130: A chromosome-level assembly is not "a complete genome".

Line 138: It shows that "Flavobacterium spp." includes multiple species. Is *Flavobacterium* sp. correct?

Lines 140-143: Were there any sequences from the algal symbionts, as shown in Supplementary Fig. 41?

Line 153: Supplementary Table 6 shows repeat contents, not gene numbers. It is

shown in Supplementary Table 10.

Lines 200-202: Were syntenic regions between sponges and ctenophores found? This comparison will be also useful to discussions on early animal evolution.

Lines 225-227: Were the genes for DNA methyltransferase any specific characters in sponges when they had highly methylated genomes?

Lines 281-286: The common gene losses may happen on transition to freshwater but it is unclear why shared gene gains were not surveyed using the same data sets. The explanation was found in S7.4 of Supplementary file 1 but more duplications in marine lineages do not exclude the possibility of specific gene gains in freshwater lineages.

Lines 414-447: The section for host-microbe associations is not likely related to the genome assembly at chromosomal level and discussions for the animal evolution.

Fig. 2d: In the chromosome figure, what is shown by black bands?

Fig. 4c: It is difficult to distinguish dots in green and blue.

Fig. 4d: Are almost all of duplicated genes found on a chromosome?

Reviewer #2:

Remarks to the Author:

In the current manuscript Kenny et al present the chromosome level assembly of the freshwater sponge *E muelleri*. This constitutes a useful resource for the scientific community interested in sponges and early animal evolution. The datasets consist of the genome sequence, synteny blocks, as well as RNA-seq and DNA methylome tracks accompanied by a useful and intuitive genome browser. The genome size, as well as gene numbers and annotations are in line with what is expected from previous work. DNA methylation patterns are similar to what has been observed in other invertebrates, and are significantly lower than those of *A. queenslandica*. The authors also provide gene expression datasets of early development and characterize the *E muelleri* microbiome. Even though the study is not particularly groundbreaking in any of its aspects, I believe that the presented analyses as well as the breadth of accompanying datasets merit publication in Nature Communications. My minor suggestions for improvement can be found below.

Page 9, line 255: "...suggesting that not all repeats are actively targeted by DNA methylation in *E. muelleri*."

Do the authors suggest that some repeats are indeed actively targeted by 5mC? Perhaps it would be interesting to analyse this in more detail. Is there anything specific about those actively-targeted repeats? Subfamily, evolutionary age?

Page 11, line 332: I found the analyses of developmental RNA-seq datasets somewhat underwhelming. I believe that these datasets allow for more exploration. Firstly it would be useful to describe in more detail the developmental stages analysed and why these stages were selected. I think that some description of how sponge developmental gene expression correlates with bilaterian and cnidarian development would be a valuable addition to the manuscript. It would be especially interesting to perform analyses of signalling pathways such as Wnt, transforming growth factor β , Notch, Hedgehog and similar pathways (and comment on their presence / absence in Ephydatia). I understand that some of this has been previously addressed, nevertheless revisiting this in light of a novel sponge genome, would be beneficial.

I would also suggest toning down the title ("The genomic basis of animal origins: a chromosomal perspective from the sponge *Ephydatia muelleri*") given that this is not the first sponge genome published and that no major novelties associated with "animal origins" are discussed.

Reviewer #3:

Remarks to the Author:

NCOMMS-20-06447

I was specifically asked to assess the microbiome analysis presented in lines 421ff and hence restrict my comments to this.

There are two aspects to the microbiome analysis. Firstly, a near complete genome of a *Flavobacterium* species was assembled from the shotgun sequencing data of a single sponge individual, and secondly a 16S rRNA gene amplicon study was performed on a number of sponge individuals collected in the Northern hemisphere.

The assembly and quality control of the *Flavobacterium* genome was done competently and shows that it likely presents a new species. The only technical comment I have here is that the 16S rRNA gene-based phylogeny presented in Supp Figure 12 looks to me like a cladogram, rather than an actual phylogram. The

labels of this figure were also impossible to read. I would have also appreciated a more in-depth comparison of this apparent sponge-associated bacterium with free-living counterparts to reveal potential adaptations to symbiosis, but understand that this might be beyond the scope of the current study. Nevertheless, please ensure that the near-complete genome is deposited in a public database and provide the relevant accession number.

The microbiome analysis was also performed overall competently and I only have some minor comments (see below). I am not sure, if the data supports the notion that the microbiomes of individual samples were "highly similar" (see Supp info page 100, last paragraph and main text line 435) given that the BC dissimilarities range from over 40-90% (see Figure S 36A). Please reconsider this conclusion. I also think it would be valuable to state in the main text (line 441ff) that the four "core" ASV were quite variable in abundance across the samples (perhaps give the range of relative abundances). This has implication for the notion given that they might be "fundamental for the metabolic function of *E. muelleri*".

Minor comments:

Supp. Info page 46: Please refer here and elsewhere to "16S rRNA gene phylogeny" instead of "16S phylogeny".

Supp info page 97: What specifically is meant when sequence "aligned poorly" to the Silva reference database.

Supp info page 97: What algorithm was used for taxonomic assignments?

Supp info page 98: Please clarify, if the read counts given and saturation results are based on the dataset where the abundant mitochondrial sequence was removed. It should.

Supp info page 100: Please make sure that writing is so that relative abundances, which are determined here, are not confused with absolute abundances.

Supp info page 100: "Archaea" not "Archaeas".

Supp info page 100, last sentence: Supp Figure 37A does not show any quantitative data on diversity and is therefore not suitable to support the statement made here.

Supp info page 102: Please provide information of what fraction of ASV sequence were used for the functional prediction, so that the accuracy can be evaluated.

Re: Kenny et al, April 28, 2020

Many thanks for your very constructive reviews of our work. Using the “track changes” feature of Word, and as noted point-by-point below in response to your suggestions, we have altered our manuscript and supplementary files in a way that we believe has completely addressed the points you have raised.

Throughout our response, we have added extra evidence, providing several new and revised figures as well as additional details in text, which have added clarity and improved our manuscript. We hope you are as happy with the revised form of this manuscript as we are.

Sincerely,

Sally Leys, on behalf of the authors, Kenny et al 2020

Please note: in response to suggestions from the community based on our preprint, we have also made the following changes to our work:

Supplementary Figure 34: Additional sequences have been added to this figure and the information provided in text has been updated. The alignments and sequence data for this figure have also been added to Supplementary File 11.

Fig 4D label: the gene name GABA has been corrected to read mGABAR. The text is changed from “Phylogeny of mGluR and GABA receptor genes, rooted with *Capsaspora owczarzaki* sequences” to: “Phylogeny of mGluR and mGABAR genes, rooted with *Capsaspora owczarzaki* sequences” (underlined = new).

We have added a figure showing GC content of scaffolds to Supplementary Figure 7, and a reference to this Figure in text (Supplementary File 1, paragraph immediately preceding this Figure).

Reviewer #1 (Remarks to the Author):

Kenny et al. have sequenced and assembled the genome of the freshwater sponge Ephydatia muelleri using a PacBio long-read sequencing, Chicago library, and Dovetail HiC library. The quality of the assembled genome seems to be at the chromosomal level, much better than available sponge genomes. They performed many analyses including DNA methylation, gene expressions from four developmental stages, and amplicon analysis of associated bacteria. The freshwater sponge genome at the chromosomal level will provide a resource on studying early diverging groups among animals. However, the authors jumped to conclusions without examining the gene expressions on adaptation to freshwater environments. In case of the Daphnia genome paper (Colbourne et al. 2010), responsive genes to hypoxia environment have been reported with differential expression data. In addition, several points need clarification.

Many thanks for your constructive comments. We have addressed the points raised by the reviewer below (see point 2 for response about gene expression, and our response to reviewer 2).

Detailed points:

Line 1: The title is attractive but there is less discussion about animal origins at the chromosomal level. Rather, they claim that the genome offers a rich resource for future study.

Reply: We considered both reviewers' comments about the title, and discussed options with our co-authors. We decided upon a minor revision to the title, removing the reference to the problematic word "origin", and replacing that with "evolution". It is now titled: "Tracing animal genomic evolution with the chromosomal-level assembly of the freshwater sponge *Ephydatia muelleri*". While we agree that we do not focus only on animal origins, our manuscript very much addresses multiple aspects of the evolution of animal genomes, including insight into the evolution of gene regulation in animals, the evolutionary implications for gene gain and loss across animal phyla, as well as analyses of gene expression and its implications for the evolution of animal traits. The word chromosomal also more precisely refers to the genome assembly. We also acknowledge reviewer 1's comment that the title was 'attractive' and feel that for a publication with broad reach and broad implications it is important for it to have an attractive title. We believe this small change to the title addresses the reviewers' concerns.

Lines 56-57: The analysis in this manuscript does not offer clarity on the adaptation to extreme temperatures and freshwater. Differential gene expression analysis or comparative genomic analysis between relative species should be provided.

Reply: The reviewer asks here, and in the first paragraph above, for experimental data on differential gene expression or comparative genomics analyses related to exposure to freshwater and saline conditions. However, as *Ephydatia* and its relatives are obligately freshwater species, we cannot generate this by exposure to other water types (in contrast with the *Daphnia* paper, where levels of hypoxia could be tested). For example, in earlier work by Adams, Goss & Leys (*PLoS One* 2010) it was found that a very slight increase in salinity (<0.1% w/v), which was required to measure conductance in the medium, reduced resistance across the freshwater sponge epithelium by 70% within 10 minutes, causing widespread damage unsuitable for measurement with a transcriptomic approach. Longer exposures kill the tissues completely. We can, however, address this point by providing detailed information about the expression of positively-adapted genes in particular, as these will reflect the changes that natural selection has found most important in *E. muelleri* adaptation (to freshwater, temperature, or other phenomena).

This proved to be very informative, and thank you for the suggestion. As the reviewer suspected, this does provide meaningful extra information, as almost all of the genes under selection in freshwater species (and *E. muelleri* in particular) are differentially expressed in the course of development. This points to specific roles for these positively selected genes in this process, which is an ideal stepping off point for future experimental studies.

We have added significantly to Supplementary File 1 to provide detail on this point. We have not copied this text here for reasons of brevity, but we have

- Added an additional element to Supplementary File 1 Figure 22,
- Added text spanning pages 56 and 57
- Made this addition obvious in the main text, by adding a note, lines 310-312:

“Almost all of the *E. muelleri* genes in these orthogroups, 85 of 117, are differentially expressed across the process of development, underlining their importance to *E. muelleri* biology (Supplementary File 1, Fig. 22E).”

While we cannot perform the reviewer’s suggested experiment, we feel this extra data makes the same point as they request.

Lines 98-101: It is unclear how the time tree was constructed in Fig. 1. Although they referred to Schuster et al. (2018), it reported the Demospongiae phylogeny and did not show the metazoan phylogeny, including the Calcarea, Homoscleromorpha, Bilateria, and so on. Supplementary Fig. 2 shows the new result of metazoan phylogenetic analysis and likely includes more broad interest result.

Reply: We apologise that this was not clearer in the text. Fig 1 is a cladogram rather than a novel phylogenetic tree, showing the time-calibrated results of Schuster et al. (2018), together with those of Dos Reis et al. (2015). This has now been made clear in the figure legend as follows:

“These estimates of divergence were taken from prior analyses as noted in Supplementary File 1, Fig 1.”

We have not used the tree shown in Supplementary Fig. 2 for two reasons. A) it is not possible to calibrate this tree, and we wish to show the time of divergence for freshwater sponges, and B) it is too complex, with extra taxa that obscure our main aim in Fig 1: to show where freshwater sponges sit within sponge phylogeny.

Line 130: A chromosome-level assembly is not “a complete genome”.

Changed to “*A chromosomal-level genome, with a higher gene content than most animals*”

Line 138: It shows that “Flavobacterium spp.” includes multiple species. Is Flavobacterium sp. correct?

Reply: Changed to sp. in other locations (particularly in Supplementary File 1). However, here we are referring to many possible Flavobacterium species, so spp is correct. We have made this clearer by adding “member of” to the text.

Lines 140-143: Were there any sequences from the algal symbionts, as shown in Supplementary Fig. 41?

Reply: No, but these are highly unlikely to be in the genome, as the tissue was grown from a colony of gemmules that live in complete darkness in the Victoria BC drinking water head tank, and which were afterwards grown in sterile conditions in the dark.

Line 153: Supplementary Table 6 shows repeat contents, not gene numbers. It is shown in Supplementary Table 10.

Thanks! Changed

Lines 200-202: Were syntenic regions between sponges and ctenophores found? This comparison will be also useful to discussions on early animal evolution.

Reply: We attempted exactly this analysis, hoping to include it in the paper, but the fragmented nature of ctenophore genomes (over 5000 scaffolds, compared to 1400 for *Ephydatia* and *Trichoplax*, and only about 800 for *Branchiostoma*, and no chromosome-length assembly) meant that no useful results could be gained – there were not enough genes on each scaffold to make any inference. We did however include comparisons to the cnidarian *Nematostella vectensis* and the placozoans *Trichoplax adhaerens* and *Hoilungia hongkongensis* as detailed in Supplementary File 1 S3.6, with more figures in Supplementary File 2.

The N50 scaffold size for *Pleurobrachia* is only 23,607 bp, enough for around one gene, and for *Mnemiopsis* it is 187,314 bp, and each scaffold contains only a handful of genes (< 5). This is not enough to detect macrosynteny and gave no useful results. It will however be a key analysis when more complete ctenophore genomes are available, adding to the 3 non-bilaterian animal phyla compared here.

Lines 225-227: Were the genes for DNA methyltransferase any specific characters in sponges when they had highly methylated genomes?

Reply: Sponges have the same set of DNA methyltransferases as other animal lineages and most of the domain architecture of these proteins is highly conserved across the animal kingdom. However, in a previous study (de Mendoza et al. 2019 Nature Ecology Evolution) it was noticed that *Amphimedon queenslandica* has a long insertion in the PWWP domain. The PWWP domain binds to histone tail modifications and helps targeting of DNMT3A/B to DNA in mammals (PMID: 25607372). Therefore, it could be that some modification in the PWWP domain somehow helps explain the hypermethylated genome of *Amphimedon*.

Consistently, the PWWP domain of *Ephydatia muelleri* DNMT3 lacks such an insertion (which is the ancestral state in animals). However, *Sycon ciliatum* also has a “canonical” PWWP domain and yet has much higher methylation levels than *Ephydatia*. Furthermore, we do not know if the insertion in *A. queenslandica* has any effect on the binding capacity of the PWWP domain. Moreover, vertebrates have “canonical” PWWP domains and hypermethylated genomes. Therefore, at this point we believe it is too speculative to link hypermethylation to a small change in the methyltransferase genes but rather consider it a response to a long evolutionary process. We have now included a PWWP domain alignment in Supplementary Figure 19 and we mention these considerations in the Supplementary material S6.2, page 44.

Lines 281-286: The common gene losses may happen on transition to freshwater but it is unclear why shared gene gains were not surveyed using the same data sets. The explanation was found in S7.4 of Supplementary file 1 but more duplications in marine lineages do not exclude the possibility of specific gene gains in freshwater lineages.

Reply: Gene gains were surveyed using the same data sets, as noted by the reviewer. This data is mapped onto Supplementary Figure 23 and is noted in Supplementary File 1 S7.4. We have added a note to the text in line 287:

“Gene gain is also noted in Supplementary File 1, S7.4, Supplementary Figure 23, although these gains are lineage specific.”

Lines 414-447: The section for host-microbe associations is not likely related to the genome assembly at chromosomal level and discussions for the animal evolution.

Reply: Host-microbe associations are, however, increasingly acknowledged to be an important feature of animal evolution, and in particular they are relevant for sponge biology because of the transparency of the associations, and as such we feel they are important and even necessary to discuss in our work. We note that the third reviewer was asked particularly to focus on this aspect of our analysis, which highlights its importance.

Fig. 2d: In the chromosome figure, what is shown by black bands?

Reply: These represent the location of the genes that show synteny on each chromosome. This information has been added to the legend as follows:

“Red boxed locations are shown in detail to the right of each matrix as chromosomal representations. Each black line on the chromosome images represents a gene linked in green to the homologous gene in the other species”

Fig. 4c: It is difficult to distinguish dots in green and blue.

Reply: These colours (green and purple) were chosen to avoid problems for the colour blind but we feel are downsampled poorly in the review pdf. In electronic format, and in the final version, these will be able to be magnified for ease of reading. We have added the high resolution figures from our paper in a variety of formats to Supplementary File 11 for ease of review and future utility.

Fig. 4d: Are almost all of duplicated genes found on a chromosome?

Reply: 127 GABAR genes are found in 31 scaffolds, but 48% of the genes appeared in 3 scaffolds (IDs: 4 (14 genes), 13 (with 26 genes), and 22 (with 21 genes)). The rest of scaffolds have from 1 to 8 genes, but most just have 1 or 2.

We have adjusted the text as follows (line 316): added “48% of which are on scaffolds 4, 13 and 22.”

Reviewer #2 (Remarks to the Author):

*In the current manuscript Kenny et al present the chromosome level assembly of the freshwater sponge *E muelleri*. This constitutes a useful resource for the scientific community interested in sponges and early animal evolution. The datasets consist of the genome sequence, synteny blocks, as well as RNA-seq and DNA methylome tracks accompanied by a useful and intuitive genome browser. The genome size, as well as gene numbers and annotations are in line with what is expected from previous work. DNA methylation patterns are similar to what has been observed in other invertebrates, and are significantly lower than those of *A. queenslandica*. The authors also provide gene expression datasets of early development and characterize the *E muelleri* microbiome. Even though the study is not particularly groundbreaking in any of its aspects, I believe that the presented analyses as well as the breadth of accompanying datasets merit publication in *Nature Communications*. My minor suggestions for improvement can be found below.*

Our thanks to the reviewer for their constructive feedback, and we hope that they find this data useful going forward.

Page 9, line 255: “...suggesting that not all repeats are actively targeted by DNA methylation in *E. muelleri*.”

Do the authors suggest that some repeats are indeed actively targeted by 5mC? Perhaps it would be interesting to analyse this in more detail. Is there anything specific about those actively-targeted repeats? Subfamily, evolutionary age?

Reply: At the reviewer’s suggestion, we have now done a more in-depth analysis on the relationship between 5mC and repeats. We have found that many LTR transposable element classes are indeed more likely to be targeted by 5mC irrespective of their genomic position (either intergenic or intronic). In contrast, DNA transposons are less likely to be highly methylated outside gene bodies. Furthermore, we have found a positive correlation between DNA methylation levels and divergence of the transposable elements. Divergence is an estimate of transposable element age; thus, we hypothesize that ancient transposons are more likely to be targeted by 5mC, specially retrotransposons with LTRs. We have now added two supplementary figures with these results in Supplementary Figure 19 and mention these observations in the main text.

Main text changes:

Added:

“In fact, repeat methylation level positively correlates with age of the repeat, and LTR retrotransposons are more likely targeted by DNA methylation irrespective of genome position (Supplementary Fig 19).”

Page 11, line 332l: I found the analyses of developmental RNA-seq datasets somewhat underwhelming. I believe that these datasets allow for more exploration. Firstly it would be useful to describe in more detail the developmental stages analysed and why these stages were selected. I think that some description of how sponge developmental gene expression correlates with bilaterian and cnidarian development would be a valuable addition to the manuscript. It would be especially interesting to perform analyses of signalling pathways such as Wnt, transforming growth factor β , Notch, Hedgehog and similar pathways (and comment on their presence / absence in Ephydatia). I understand that some of this has been previously addressed, nevertheless revisiting this in light of a novel sponge genome, would be beneficial.

Reply: This analysis is already in progress in some detail for a follow-up manuscript, which will be dedicated to a more focussed investigation on development than was possible here due to space. However, we recognise and understand the value of adding extra detail to this paper as a resource in the interim. To address the reviewer’s comments we have added two extra sections to our supplementary document, (9.4 and to S11) together comprising approximately five pages of notes and analysis of results, as well as a new introduction to section 8.1 which also adds an extra depth of detail. We have also conducted additional analyses, and present these in a new Supplementary Figure 36, detailing expression of the *Wnt*, *transforming growth factor β* , and *Hedgehog* pathways across the process of development using our RNAseq data.

These sections have added

- Extra detail about our experiment and findings (section 8.1, page 78)

- Detailed discussion of the *Wnt*, *transforming growth factor β* , and *Hedgehog* pathways (Supp Fig 36) their expression in the course of development, and how this compares to other bilaterian and cnidarian models (section 9.4, pages 91-94).
- Considerable extra data on the process of development in *E. muelleri* (S11, pages 111-113)
- Notes calling attention to this data in the main text (e.g. lines 367-376)

We have not copied this extra information here for reasons of brevity, but this considerable extra detail and analysis should comprehensively address the reviewer's suggestions.

*I would also suggest toning down the title ("The genomic basis of animal origins: a chromosomal perspective from the sponge *Ephydatia muelleri*") given that this is not the first sponge genome published and that no major novelties associated with "animal origins" are discussed.*

Reply: We have revised the title in response to the reviewer's suggestion, and that of reviewer 1. It is now titled: "Tracing animal genomic evolution with the chromosomal-level assembly of the freshwater sponge *Ephydatia muelleri*"

Reviewer #3 (Remarks to the Author):

I was specifically asked to assess the microbiome analysis presented in lines 421ff and hence restrict my comments to this.

*There are two aspects to the microbiome analysis. Firstly, a near complete genome of a *Flavobacterium* species was assembled from the shotgun sequencing data of a single sponge individual, and secondly a 16S rRNA gene amplicon study was performed on a number of sponge individuals collected in the Northern hemisphere.*

*The assembly and quality control of the *Flavobacterium* genome was done competently and shows that it likely presents a new species. The only technical comment I have here is that the 16S rRNA gene-based phylogeny presented in Supp Figure 12 looks to me like a cladogram, rather than an actual phylogram. The labels of this figure were also impossible to read. I would have also appreciated a more in-depth comparison of this apparent sponge-associated bacterium with free-living counterparts to reveal potential adaptations to symbiosis, but understand that this might be beyond the scope of the current study. Nevertheless, please ensure that the near-complete genome is deposited in a public database and provide the relevant accession number.*

Many thanks to the reviewer for their comments and expert advice. In Supp Figure 12 we have inserted the word "cladogram", and the figure is available for download in high definition from Supplementary File 11 and <https://spaces.facsci.ualberta.ca/ephybase/>, and a note to this effect has been added to the legend.

The accession number has now been added to the text (Line 552):

"This project has been deposited at DDBJ/ENA/GenBank under the accession JABACO000000000. The version described in this paper is version JABACO010000000. The sequence of *Flavobacterium* sp. has also been uploaded, with accession number CP051546."

The microbiome analysis was also performed overall competently and I only have some minor comments (see below). I am not sure, if the data supports the notion that the microbiomes of individual samples were “highly similar” (see Supp info page 100, last paragraph and main text line 435) given that the BC dissimilarities range from over 40-90% (see Figure S 36A). Please reconsider this conclusion.

Reply: The text has been modified as suggested. Line 435 (now line 444) has changed to read "Overall, differences in microbiome content were determined by geographic location, as has been found in marine sponges". The supplementary text has been altered to read “the microbial community was most well-grouped within each sampling site”

I also think it would be valuable to state in the main text (line 441ff) that the four “core” ASV were quite variable in abundance across the samples (perhaps give the range of relative abundances). This has implication for the notion given that they might be “fundamental for the metabolic function of E. muelleri”

Reply: The text has been corrected, and now reads: (Line 449 in corrected text) “we found that four amplicon sequence variants were shared among all samples, yet with different percentages ranging from < 1% to > 20% in different samples.”

Supp. Info page 46: Please refer here and elsewhere to “16S rRNA gene phylogeny” instead of “16S phylogeny”.

Reply: Replaced in 5 locations

Supp info page 97: What specifically is meant when sequence “aligned poorly” to the Silva reference database.

Reply: By this phrase we meant ‘as assessed by the mothur MiSeq SOP pipeline’. During the alignment with the Silva database, some of the sequences are eliminated with a warning saying " X of your sequences generated alignments that eliminated too many bases, a list is provided in file.trim.contigs.pcr.good.unique.flip.accnos". Our sentence referred to that filtering step in the pipeline. We have added “assessed by the mothur MiSeq SOP pipeline as generating alignments that eliminated too many bases” to the supplementary text, page 101, to make this clearer.

Supp info page 97: What algorithm was used for taxonomic assignments?

Reply: The algorithm used is the one included in the mothur pipeline (<https://mothur.org/wiki/classify.seqs/>), when using the "classify.seqs" command. This algorithm is specifically a Bayesian naive classifier (Naive Bayesian classifier for rapid assignment of rRNA sequences into the new bacterial taxonomy. Wang Q, Garrity GM, Tiedje JM, Cole JR. Appl Environ Microbiol. Aug;73(16):5261-7. 2007).

This information has been given now on page 97.

Supp info page 98: Please clarify, if the read counts given and saturation results are based on the dataset where the abundant mitochondrial sequence was removed. It should.

Reply: The reviewer refers to the very first description of the dataset, with all sequences included. We have changed the text as follows below, to add the number of read counts after

the removal of unknown sequences, and make it clear that mitochondrial sequence was removed, as follows:

“During the taxonomic annotation, we noticed that between 18.8 and 84.2% of raw counts were annotated as “unknown”, which was larger than expected. In fact, only one unknown ASVs was dominant in all samples (>97% of the unknown counts). A Blast similarity search of this ASV on NCBI matched with *Ephydatia muelleri* genome assembly, organelle: mitochondrion (100% sequence identity, Genbank: LT158504.1). This ASV, and other unknown ASVs were excluded from further analysis, leaving a filtered dataset with 12,922 to 65,258 raw counts, which showed saturation in rarefaction curves (Supplementary File 10: 10A), and between 248 to 4,061 unique ASVs.”

Also, the new saturation curves do not change our results, but we have substituted the graph in Supplementary File 10.

Supp info page 100: Please make sure that writing is so that relative abundances, which are determined here, are not confused with absolute abundances.

Reply: Added in 3 locations in text on this page

Supp info page 100: “Archaea” not “Archaeas”.

Reply: Changed

Supp info page 100, last sentence: Supp Figure 37A does not show any quantitative data on diversity and is therefore not suitable to support the statement made here.

Reply: Thank you for catching that. We meant 37B not 37A, although please note that due to the insertion of a new figure, this is now 38B. The sentence "Samples from Sooke reservoir included unhatched and hatched gemmules, without large differences among them." was supported by Figure 37B (now 38B) (the PCoA plot) where there is complete overlap of unhatched and hatched gemmules. We have moved the sentence to its correct location, under the Beta diversity paragraph, and modified it for clarification.

Supp info page 102: Please provide information of what fraction of ASV sequence were used for the functional prediction, so that the accuracy can be evaluated.

Reply: We apologise for leaving these out. The % of the sequences UNUSED in the prediction ranged from 0.14% to 0.83%. These have been provided on page 106.

Specific values are:

Montgomery Canal Ind 1 adult (a) 0.28743

Montgomery Canal Ind 1 adult (b) 0.45442

Montgomery Canal Ind 2 adult (a) 0.33108

Montgomery Canal Ind 2 adult (b) 0.28994

Montgomery Canal Ind 3 adult (a) 0.80292

Montgomery Canal Ind 3 adult (b) 0.8281

Youngs Pond hatched (a) 0.33143

Youngs Pond hatched (b) 0.26188

Sooke Reservoir Ind 1 unhatched (a) 0.59985

Sooke Reservoir Ind 1 unhatched (b) 0.55047

Androscoggin River hatched (a) 0.23593
Androscoggin River hatched (b) 0.24604
Twitchell Brook hatched (a) 0.18642
Twitchell Brook hatched (b) 0.16903
O'Connor unhatched (a) 0.23265
O'Connor unhatched (b) 0.14755
O'Connor hatched (a) 0.44069
O'Connor hatched (b) 0.37871
Sooke Reservoir Ind 2 unhatched (a) 0.71708
Sooke Reservoir Ind 2 unhatched (b) 0.75985
Sooke Reservoir Ind 3 hatched 5 Mar 0.71608
Sooke Reservoir Ind 3 hatched 15 Mar 0.5216

We contacted the authors of Tax4Fun before we submitted the manuscript, because we thought that these values were not optimal, but we did not obtain a clear explanation. In fact, using their example dataset unused values were always above 0.90%, so we believe that our prediction was acceptable.

Reviewers' Comments:

Reviewer #1:

Remarks to the Author:

The authors revised to almost all of comments properly. One concern is about the revised sentence in abstract (lines 62 - 64). I feel the possible relationship between environmental adaptation and duplicated genes could be discussed in the manuscript since I cannot find it in the main text. Do 85 orthologroups with differential expressions (lines 332 - 334) include large number of lineage-specific duplicated genes since a gemmule stage is required for adaptation to extreme habitat?

Reviewer #2:

Remarks to the Author:

The authors have done a good job in addressing my queries (as well as those of other reviewers). In my opinion the manuscript can be published in Nature Communications.

Kenny et al. Tracing animal genomic evolution with the chromosomal-level assembly of the freshwater sponge *Ephydatia muelleri*

We are grateful for the comments and suggestions of the reviewers, and believe we have been able to modify the manuscript accordingly.

Responses to the reviewers:

“Reviewer #1 (Remarks to the Author):

Reviewer: The authors revised to almost all of comments properly. One concern is about the revised sentence in abstract (lines 62 - 64).

Response: We have changed largely to often, and also changed show to suggest, both of which we think support the data and maintains the concept, as well as softens the sentence.

“Our analyses reveal a metazoan-typical genome architecture, with highly shared synteny across Metazoa, and suggest that adaptation to the extreme temperatures and conditions found in freshwater often involves gene duplication.”

Reviewer: I feel the possible relationship between environmental adaptation and duplicated genes could be discussed in the manuscript since I cannot find it in the main text.

Response: We have added a reference to this in the introduction (line 287) and address this in the main text lines 361 to 371. We added new text regarding adaptation and gene duplication on lines 493 to 512.

Reviewer: Do 85 orthogroups with differential expressions (lines 332 - 334) include large number of lineage-specific duplicated genes since a gemmule stage is required for adaptation to extreme habitat?

Response: It is true that we don't know whether some of the 85 genes are involved in gemmule formation, partly because gemmules are simply excessively well produced collagenous structures and so presumably use many similar components the adult (hatched) sponge uses in its filter feeding life. But these genes are expressed during development, which is important to the long-term success of the organism. We have illustrated this with examples that support the extreme duplications of these genes through the different stages of development on Lines 536 to 563 as follows:

“Almost all of the *E. muelleri* genes in these orthogroups, 85 of 117, are differentially expressed across the process of development, underlining their importance to *E. muelleri* biology (Supplementary Fig. 22E). It is not uncommon for these differentially expressed genes to have multiple in-paralogs. The most prolifically duplicated genes are a *cytoplasmic actin* and a *leukotriene receptor* which are both tandemly duplicated 6 times in the *E. muelleri* genome, from loci Em0009g1201 and Em0009g943 respectively. Altogether, of the 85 differentially expressed, positively selected orthogroups, 54 are single copy, and 31 possess two or more in-paralogs of the genes tested (Supplementary Data 6). This indicates that duplication of these genes is commonly associated with adaptation.”

Reviewer #2 (Remarks to the Author):

The authors have done a good job in addressing my queries (as well as those of other reviewers). In my opinion the manuscript can be published in Nature Communications.

No responses.

We have also updated citation 33 to a more recent and appropriate paper.

Sincerely,
Sally Leys & Nathan Kenny